# Dynamic Crushing Analysis of a Three-Dimensional Re-Entrant Auxetic Cellular Structure

**DOI:** 10.3390/ma12030460

**Published:** 2019-02-01

**Authors:** Tao Wang, Zhen Li, Liangmo Wang, Zhengdong Ma, Gregory M. Hulbert

**Affiliations:** 1Department of Mechanical Engineering, Nanjing University of Science and Technology, Nanjing 210094, China; zhen_li@njust.edu.cn (Z.L.); liangmo@njust.edu.cn (L.W.); 2Department of Mechanical Engineering, University of Michigan, Ann Arbor, MI 48109, USA; mazd@umich.edu (Z.M.); Hulbert@umich.edu (G.M.H.)

**Keywords:** 3D auxetic structure, crushing mode, dynamic plateau stress, energy absorption, finite element simulation

## Abstract

Dynamic behaviors of the three-dimensional re-entrant auxetic cellular structure have been investigated by performing beam-based crushing simulation. Detailed deformation process subjected to various crushing velocities has been described, where three specific crushing modes have been identified with respect to the crushing velocity and the relative density. The crushing strength of the 3D re-entrant auxetic structure reveals to increase with increasing crushing velocity and relative density. Moreover, an analytical formula of dynamic plateau stress has been deduced, which has been validated to present theoretical predictions agreeing well with simulation results. By establishing an analytical model, the role of relative density on the energy absorption capacity of the 3D re-entrant auxetic structure has been further studied. The results indicate that the specific plastic energy dissipation is increased by increasing the relative density, while the normalized plastic energy dissipation has an opposite sensitivity to the relative density when the crushing velocity exceeds the critical transition velocity. The study in this work can provide insights into the dynamic property of the 3D re-entrant auxetic structure and provides an extensive reference for the crushing resistance design of the auxetic structure.

## 1. Introduction

Auxetic cellular structures have attracted great attention since, unlike conventional honeycomb, they expand laterally when stretched and contract laterally when compressed, featuring a negative Poisson’s ratio (NPR). The termed ‘auxetic effect’ brings comprehensive improvements of mechanical properties, like increased stiffness, high shear modulus, indentation resistance and energy absorbing performance, to the cellular structure. These fascinating properties all contribute to augmenting their applications in automotive, packaging and structural protection [1,2,3]. In recent years, increasing interests have been devoted in this field and the research results can be summarized to cover the topology design, property investigation, rapid prototyping and utilization of the auxetic structures [4,5,6,7,8]. 

Due to superior performance in dissipating energy, auxetic cellular structures can work as a better alternative in the design of sandwich structure and vehicle crash box, which illustrates the significance of dynamic crushing analysis of auxetic structures [9,10]. To investigate the dynamic properties of auxetic cellular structures, a few studies of theoretical models and experiments have been conducted and some remarkable conclusions have been demonstrated. Grujicic et al. [11] have investigated the dynamic response of the auxetic sandwich structures by using a multiphysics computational approach. The major deformation modes of the structures subjected to blast loading have been highlighted in the paper. The performance of an auxetic re-entrant honeycomb-cored sandwich panels under close-in blast loadings has been compared with that of the conventional steel honeycomb panels of the same size [12]. Both the simulations and tests validated the substantial contribution of auxetic characteristics in improving the force mitigation and blast-resistance performances of auxetic sandwich panels. The auxetic core microstructure has significant influence in the crushing resistance, as the collapse of auxetic cell is highly related to the crushing strength of cellular structure [13]. The enhancement of the dynamic responses and blast resistance of the sandwich structures with auxetic re-entrant cell honeycomb cores under blast loading has also been concluded in these studies [14]. To further explore the dynamic property of the auxetic honeycomb subject to large deformation, Wangyu Liu et al. [15] studied the characteristics of the peculiar crushing patterns and the energy absorption performances of the re-entrant hexagonal honeycomb. Their studies pointed out that the auxetic effects determines the impact response of the re-entrant honeycomb by affecting the crushing pattern. A similar conclusion was also reported by Xiuhui Hou et al. [16], who attributed the superior impact resistance of auxetic honeycombs to the auxetic effect induced in the re-entrant topology.

However, it can be concluded from the reviews that most recent research has focused on the two-dimensional honeycombs. Two-dimensional auxetic honeycomb possesses NPR only in two in-plane orthogonal directions (the out-of-plane does not exhibit auxetic behavior), while three-dimensional (3D) auxetic cellular structure can easily achieve isotropic NPR in three principle directions. Considering the poor predictability of virtual crushing directions, 3D auxetic cellular structure may be more suitable in many application scenarios, thus deserving in-depth investigation.

In this paper, a three-dimensional auxetic cellular structure consisting of re-entrant hexagonal cells, which is illustrated in Figure 1, has been modeled and analyzed in the dynamic crushing mechanics. As a typical NPR configuration, this design was originally proposed by Evans [17] and has been numerically and empirically studied by other researchers [18,19,20,21]. Though these studies are limited in analytically modeling and elastic analysis, some conclusions can be seriously referred when extending the research towards dynamic crushing domain. With numerical simulations, the dynamic crushing analysis of the 3D re-entrant cellular structure has been conducted in the variable strain rates ranging from 5 s^−1^ to 120 s^−1^. The dynamic behaviors of the auxetic structure, including the crushing modes, the crushing strength and the energy absorption capacity, have been further investigated with respect to the crushing velocity and the structure relative density.

## 2. Geometry and Modeling

### 2.1. Geometry

Figure 1a depicts the employed 3D re-entrant auxetic cellular structure, which is arrayed by repeating the representative volume element (RVE) shown in Figure 1b in all three principle directions. As can be seen from Figure 1b, the RVE can be described with a pair of orthogonally arranged re-entrant hexagons, which consist of several vertical and inclined struts. By referring to the previous publications, the struts are assumed to have the identical square-shape cross-sections. Figure 1c illustrates the whole view of the geometrical parameters for characterizing the RVE. The lengths of the inclined struts and vertical struts are defined as L and H, respectively. The thickness of the strut cross-section is defined as t. The re-entrant angle between the vertical and inclined struts is defined as θ. Therefore, the geometry of the RVE can be determined independently with four parameters namely L, H, t and θ. Noting that to ensure re-entrant configuration, a geometric constraint H>2Lcosθ should be satisfied in initial design. Considering the symmetry of the 3D structure, it is apparent that the mechanical properties along the x and y directions are of the same.

The materials of the struts may be varied to achieve different properties. As a preferred embodiment, they are assumed to have the same material in this study. Then, as shown in Figure 1, the relative density of the 3D re-entrant auxetic structure, $\rho_{R}$, that is, the ratio between $\rho_{0}$ the effective density of cellular structure and $\rho_{s}$ the density of strut material, can be obtained by dividing the volume occupied by all the struts and the total volume. The volume occupied by all the struts is specified as $V_{s}=2Ht^{2}/2+4Ht^{2}/2+8Lt^{2}$. The total volume is calculated as $V_{0}={\left(2L\sin\theta +2t\right)}^{2}\cdot \left(2H-2L\cos\theta \right)$. Combining the independent parameters defined above, the relative density $\rho_{R}$ is therefore described as:(1)$$\rho_{R}=\frac{\rho_{0}}{\rho_{s}}=\frac{V_{s}}{V_{0}}=\frac{\left(8L+3H+2t\right)t^{2}}{{\left(2L\sin\theta +2t\right)}^{2}\left(2H-2L\cos\theta +t\right)}$$

### 2.2. Description of Finite Element Modeling

In this section, a full size finite element (FE) model of 3D re-entrant auxetic cellular structure for dynamic crushing simulation has been built using Abaqus^®^, as shown in Figure 2a. The 3D re-entrant auxetic structure adopted for simulation have been given with 12 × 9 × 6 representative cells (twelve unit cells in x-direction, nine unit cells in y-direction and six unit cells in z-direction). The geometrical parameters of the 3D specimens in crushing analysis are set as L=20 mm, H=30 mm, θ=60° and t varies from 1 mm to 4 mm, featuring a variable relative density closely between 1% and 10%. The dimensions of whole model are then given as 416 mm by 360 mm by 208 mm in Figure 2a.

In meshing of the model, by referring to the publication [22], the two-node beam element BEAM188 has been used to mesh the struts of the structure as this element type can ensure high accuracy at a low computational cost. As demonstrated in the paper [22], this element type is suitable to simulate the linear small-deflection and non-linear large-deformation of the strut, which both occur in dynamic crushing. Figure 2a also shows an example of the FE beam model of a half unit cell. As can be observed from the figure, the mesh model is constructed based on the design scheme shown in Figure 1c (red dash), as the schematic lines of beam element are directly assigned along the central axes of the struts. And all the corners of struts are defined as default nodes before meshing. As a preferred material for additive manufacturing, PA6 has been adopted as the matrix material of strut and been assumed to be a rate-independent elastic-perfectly plastic model with mass density $\rho_{s}=1.14\;{\mathrm{g/cm}}^{3}$, Young’s modulus $E_{s}=2.32\;\mathrm{GPa}$, yield stress $\sigma_{y}=115\;\mathrm{MPa}$ and Poisson’s ratio $\nu_{s}=0.34$ [23].

The detailed boundary conditions for crushing simulations are depicted in Figure 2b. To simulate the dynamic crushing process, a rigid-wall has been modeled to impact the upper cells of the specimen with a constant velocity, $v_{c}$, along the y-direction, as shown in Figure 2b. The crushing velocity $v_{c}$ varies from 2 m/s to 50 m/s in this study. All the nodes at the bottom boundary have been fixed along the y-direction as $u_{y}=0$. To fix the model, a central node at the bottom boundary has been fixed by constraining all degrees of freedom. Nodes at the lateral boundaries have been set with the coupled boundaries so as to avoid the influence of boundary effect [24]. An automatic single contact algorithm in Abaqus^®^ has been adopted to model the contact between surfaces with a friction coefficient of 0.35.

A mesh sensitivity investigation has been carried out to determine the proper mesh arrangement. The basic element size in meshing the strut varies from 4 mm to 10 mm for different model complexities. The different models are investigated under crushing at a constant velocity $v_{c}$ of 30 m/s. The convergence of the simulations is depicted using crushing reaction force in Figure 3a and hourglass error in Figure 3b.

Abaqus explicit analysis performed in the present study uses hourglass control mode. Once the artificial strain energy is greater than approximately 5% of the total energy, that is, hourglass error exceeds 5%, hourglassing could be a problem and should be further analyzed by refining mesh [25]. It can be concluded from Figure 3 that the hourglass error of meshing with basic element size of 6 mm can be limited in 5% (in Figure 3b) and is able to achieve a similar result compared with the other refined models (in Figure 3a). Thus, combining the accuracy and computational cost, the basic element size in meshing the model has been set as 6 mm for all simulation in the present work.

## 3. Dynamic Crushing Modes

As indicated in previous publications [26,27,28], the crushing modes for regular hexagonal honeycombs with same relative density are highly dependent on the crushing velocity. Inspired by this work, keeping a constant $\rho_{R}=1\%$, dynamic crushing simulation of the 3D re-entrant auxetic cellular structure has been performed under a wide range of crushing velocity $v_{c}$ varying from 2 m/s to 50 m/s. By defining the y-directional strain rate as the ratio of $v_{c}$ to the height of the structure $h_{y}$, the strain rate varies approximately from 5 s^−1^ to 120 s^−1^ in this study. Figure 4, Figure 5, Figure 6 and Figure 7 depict the deformation processes of the auxetic structures subjected to dynamic crushing along the y-direction. As seen from the simulation results, similar to the studies for regular hexagonal honeycombs [26], three specific crushing modes have also been observed from the numerical simulations for auxetic structures, namely quasi-static, transition and dynamic mode. But due to the auxetic effect, these specific crushing modes all exhibit different deformation shapes compared to the crushing patterns for regular hexagonal honeycombs reported as ‘X,’ ‘V’ and ‘I’ types [26]. Figure 4, Figure 5 and Figure 6 show the deformation processes under the crushing velocities of 5, 15 and 50 m/s, representing the three specific crushing modes, respectively. Each process recorded is given with six typical deformation stages with the longitudinal $\varepsilon_{y}$= 0.05, 0.1, 0.2, 0.3, 0.4 and 0.5.

As shown in Figure 4, when the auxetic structure is crushed under a low velocity ($v_{c}$ = 5 m/s), localized deformation occurs as the cells collapse layer by layer and obvious contraction is observed along the lateral plane. The first collapse band is initiated randomly at the 5th layer. As the crushing continues, the cells near the proximal end gradually pile up towards the collapse band, forming a densification core. Until all the cells above the collapse band densified, the densification core keeps spreading towards the distal end. No ‘X’ or ‘V’ type modes as for regular hexagonal honeycombs are observed in crushing case for the auxetic structure. Since the lateral contraction around the densification core is larger than those at the proximal and distal ends, the compressed structure shows a necking shape, which is seen as the diagnostic feature of quasi-static mode. Moreover, simulations for low-velocity cases show an instability in the growing of densification core, as shown in Figure 4. The possible cause for this phenomenon can be attributed to the reduction of cross-section area in densification core, leading to a dislocation between the densified layers.

When the crushing velocity is increased ($v_{c}$ = 15 m/s in Figure 5), the auxetic structure deforms in the same way as in the low-velocity crushing cases through accumulation of collapse layers into the densification core. But as can be seen from the figure, there is an obvious difference that the initial collapse band is generated in the layer close to the proximal end. Additionally, the lateral contractions at the proximal and distal ends almost occur simultaneously. As the crushing continues, the growing of the densification core at the upper structure and the contraction of the both ends are the major deformation pattern for the auxetic structure, featuring a spindle shape for the transition mode.

When the auxetic structure is subjected to a high-velocity crushing ($v_{c}\;$= 50 m/s in Figure 6), a shockwave like crushing front is generated at the proximal end only and then spreads towards the distal end, which is similar to the deformation pattern for regular hexagonal honeycombs in dynamic mode. Upon crushed, the cells near the proximal end rapidly collapse into a densification core and a localized contraction is also expected around the densification core, exhibiting a top-shrinking shape as a whole. In this crushing mode, no obvious collapse and contraction are observed near the distal end. This might be because that most of the crushing energy is absorbed through plastic collapse of cells in the densification core. Hence, the stress waves travelled to the distal ends are not intense enough to collapse the cells at this area [29]. This phenomenon can be further investigated by integrating with the crushing strength analysis given below.

The detailed crushing processes for auxetic structures with respect to the analyzed range of crushing velocities are shown in Figure 7, which clearly illustrates the gradual transition among three specific crushing modes. As described above, the main differences among the three crushing modes lie in the location of initial collapse band (marked in Figure 7 with red box) and the growth process of densification core. In addition to the crushing velocity, the crushing modes of auxetic structures are highly affected by the relative density. Therefore, auxetic structures with variable relative densities have been modeled by changing the thickness of strut and used to perform crushing simulations with respect to different crushing velocities.

In Figure 8, the corresponding crushing mode for each simulation case has been identified with a solid marker, forming a mode classification map for the 3D re-entrant auxetic structure. As depicted in the figure with the dashed lines, an approximately linear relativity can be recorded between the critical transition velocity, $v_{t}$ and the relative density, showing an increase of $v_{t}$ with increasing $\rho_{R}$. This can be understood by referring to the shock propagation research of cellular system, where there is a widely accepted one-dimensional shock theory using rigid, perfectly plastic, locking (R-PP-L) idealization model, revealing the dependence of critical velocity on the density of the structure [30,31,32]. By employing this theory, a similar conclusion for the regular hexagonal honeycomb has been drawn in previous studies [26,27] and the transition velocity among different modes has been further expressed intuitively [28,31].

## 4. Dynamic Crushing Strength

To further investigate the dynamic response of auxetic structure, the crushing stresses of auxetic structure ($\rho_{R}=1\%$) produced at the proximal and distal ends under different crushing velocities are given in Figure 9a,b with respect to compression strain. All the nominal stress-strain curves show an overall piecewise characteristic during crushing, featuring with the ephemeral elasticity, long plateau and final densification periods. Another characteristic is that the stress-strain curves are observed with high oscillations. The possible cause can be attributed to the localization as the cells collapse layer by layer, resulting in a sudden drop of the stress when an adjacent layer of cells starts to collapse. Thus, the fluctuations in curves are therefore expected, especially in the dynamic crushing mode ($v_{c}$ = 50 m/s).

As the crushing velocity increases, the crushing stress at the proximal end increases remarkably, while the stress at the distal end has less dependence on the crushing velocity. Additionally, the crushing stress at the proximal end is higher than that at the distal end. Since the rate-independent elastic-perfectly plastic model has been adopted and the effect of strain-rate is ignored in this study, the reason for the enhanced strength can be reasonably attributed to the change of deformation mode of struts as discussed above. The above two claims indicate that the analyzed re-entrant auxetic structure has great potential in protecting the contacted part at the distal end.

Densification strain is defined as a strain at which densification begins. Another trend in Figure 9a,b is observed that the stress-strain curves at both ends show a lengthened plateau with increasing the crushing velocity, increasing the strain at which the final densification period starts, which indicates an crushing velocity dependent densification strain in auxetic cellular structures. By referring to the publication [33], the dynamic densification strain $\varepsilon_{D}$ is defined as the intersection of the tangents to the plateau and densification regimes in this study and the value is then determined numerically from the obtained stress-strain curves within FE simulation. Figure 9c depicts the dynamic densification strains $\varepsilon_{D}$ with respect to different crushing velocities. Obviously, the dynamic densification strain increases gradually with increasing the crushing velocity and converges to a peak value when the velocity reaches around 35 m/s, showing an increase of about 10% compared with that for the quasi-static case. In this study, the dynamic plateau stress is defined as the average value of compression stresses for a compressive strain of between the 0.5% strain and the dynamic densification strain $\varepsilon_{D}$ [30]. Since in the R-PP-L idealization model [31,32], the densification strain is adopted as a rate-independent value when formulating the explicit form of dynamic plateau stress. The trend revealed in Figure 9c indicates an amendment needed to determine the dynamic plateau stress with rate sensitivity being considered, which is further discussed in the next section.

Further effect of the relative density has been analyzed, as shown in Figure 9d. As the crushing strength with high strain rate is of high interest in crashworthiness design, the density related research at the proximal end is performed within the dynamic crushing mode, where the auxetic structures with different relative densities are crushed with the same velocity, $v_{c}$ = 50 m/s. From the shown stress-strain curves, the peak stress and plateau stress increase with the relative density. As the variation of relative density in this study is achieved by varying the strut thickness, the thickened strut can present a higher resistance to collapse loading, increasing the crushing strength of cellular structure. Although a long plateau with high stress level can guarantee a satisfactory energy absorption performance, the increased relative density simultaneously generates a higher peak stress, which is not appropriate in crashworthiness design. Thus, the function-oriented design for auxetic structure deserves a more detailed guideline, not merely increasing the relative density and this is not included in the present study.

As analyzed above, the theoretical prediction of dynamic plateau stress is highly related to the crushing velocity and relative density. Ideally, in the shock theory proposed by Reid, assuming the cellular structure as the R-PP-L model, the dynamic plateau stress $\sigma_{D}$, at the proximal end is given in a general form as in Reference [30]:(2)$$\sigma_{D}=\sigma_{0}+\frac{\rho_{0}v_{c}^{2}}{\varepsilon_{D}}$$
where $\sigma_{0}$ is defined as the quasi-static plateau stress, $\varepsilon_{D}$ is the densification strain, $\rho_{0}$ is the effective density of cellular structure and $v_{c}$ represents the crushing velocity. For the quasi-static plateau stress $\sigma_{0}$ and the densification strain $\varepsilon_{D}$, Gibson and Ashby [34] formulated them as a function of the relative density $\rho_{0}/\rho_{s}$ and the yield stress of strut material $\sigma_{y}$ as below:(3)$$\sigma_{0}=C{\left(\frac{\rho_{0}}{\rho_{s}}\right)}^{2}\sigma_{y}$$
(4)$$\varepsilon_{D}=1-\gamma \frac{\rho_{0}}{\rho_{s}}$$
where $\rho_{s}$ is the density of strut material and C and γ are dimensionless constants. The marks shown in Figure 10 are the simulation results of quasi-static plateau stress $\sigma_{0}$ and densification strain $\varepsilon_{D}$ for the structures with different relative densities. By employing Equations (3) and (4), two fitting curves have been drawn on basis of the discrete marks, where the dimensionless constants of equations are derived as: C = 1.83 and γ = 0.85. The simulation results for dynamic plateau stress of auxetic structures with different combinations of crushing velocities and relative density are listed in Table 1, where the theoretical predictions using the Equations (2)–(4) are also listed as a contrast. As shown in the table, with increasing crushing velocity, an increasing overestimation amounting to about 13% has been observed for the theoretical results above the simulation results. The deviation can be reasonably attributed to the constant assumption of $\varepsilon_{D}$ in the R-PP-L model, which is actually a rate-dependent coefficient [35]. Thus, to improve the prediction accuracy of the theoretical model, a modification has been performed within the empirical formula of dynamic plateau stress.

As revealed in Figure 9c, the dynamic densification strain can be related to the crushing velocity and thus a piecewise-defined function has been formulated herein by two sub-functions, a linear function fitting for all values of $v_{c}$ less than the wave trapping speed $v_{w}$ [30] and a constant function fitting for the values of $v_{c}$ greater than or equal to $v_{w}$. The formulated piecewise-defined function is given as:(5)$$\varepsilon_{D}=\{\begin{array}{c} \left(0.1\frac{v_{c}}{v_{w}}+1\right)\left(1-\gamma \frac{\rho_{0}}{\rho_{s}}\right)\;0<v_{c}<v_{w}\\ 1.1\left(1-\gamma \frac{\rho_{0}}{\rho_{s}}\right)\;v_{c}\geq v_{w} \end{array}$$

Substituting Equations (3) and (5) into Equation (2), the dynamic plateau stress $\sigma_{D}$ considering rate-dependent $\varepsilon_{D}$ is given as:(6)$$\sigma_{D}=\{\begin{array}{c} C{\left(\frac{\rho_{0}}{\rho_{s}}\right)}^{2}\sigma_{y}+\frac{\rho_{0}v_{c}^{2}}{\left(0.1\frac{v_{c}}{v_{w}}+1\right)\left(1-\gamma \frac{\rho_{0}}{\rho_{s}}\right)}\;\;0<v_{c}<v_{w}\\ C{\left(\frac{\rho_{0}}{\rho_{s}}\right)}^{2}\sigma_{y}+\frac{\rho_{0}v_{c}^{2}}{1.1\left(1-\gamma \frac{\rho_{0}}{\rho_{s}}\right)}\;v_{c}\geq v_{w} \end{array}$$

Figure 11 shows the comparison between the simulation results and the theoretical predictions calculated following Equation (6) as for dynamic plateau stress $\sigma_{D}$. As clearly shown in the figure, for the given range of relative densities and velocities, the theoretical model with Equation (6) agrees well with the FE model with a maximum deviation of 4.5%. Considering the slight error due to the approximate expression of $\varepsilon_{D}$, the comparison has validated the correlations given in Equation (6). Therefore, the modified formula of dynamic plateau stress $\sigma_{D}$ provides a good guidance for predicting crushing strength of auxetic structure in crashworthiness design.

## 5. Energy Absorption Capacity

To explore the effects of crushing velocity and relative density on the energy absorption capacity of the re-entrant auxetic structure ($\rho_{R}=8\%$), the plastic energy dissipated $U_{p}$ during the crushing process has been monitored and plotted against the overall compression strain, as shown in Figure 12. The frictional dissipated energy $U_{f}$ due to friction between the contacted struts has also been plotted in the figure. 

As can be seen, the frictional dissipated energy accounts for a small proportion of total internal energy and the major energy absorption mechanism is dominated by the plastic bending and axial compression of struts [36]. The plotted curves show that all the plastic energy dissipations under different crushing velocities feature an almost linearly increasing trend with the increase of compression strain. In addition, the plastic energy dissipation in the same compression strain increases remarkably as the crushing velocity increases, especially for higher velocities. This might be because of the increased crushing strength of structure under high crushing velocity.

To evaluate the energy absorption efficiency from a standardized perspective, two indices have been defined by involving the quality of structure and the quasi-static dissipation energy, giving the specific plastic energy dissipation $U_{p}^{s}$ and the normalized plastic energy dissipation ${\bar{U}}_{p}$ [37], respectively. They are given herein as
(7)$$U_{p}^{s}=\frac{U_{p}}{m}$$
and
(8)$${\bar{U}}_{p}=\frac{U_{p}}{\sigma_{yc}Ah_{z}}$$
respectively, where $m=\rho_{0}V_{0}=\rho_{0}Ah_{z}$ is the mass of cellular structure, A is the effective cross-section area, $h_{z}$ is the effective height along crushing direction and $\sigma_{yc}$ is the effective yield strength of cellular structure and can be calculated as $\sigma_{yc}=0.5\rho_{R}^{2}\sigma_{y}$. According to the shock wave theory, for the continuum-based R-PP-L model, the plastic energy dissipation can be explicitly written as
(9)$$U_{p}=\sigma_{0}\varepsilon_{D}+\frac{1}{2}\rho_{0}v_{c}^{2}$$

Applying the rate-dependent $\varepsilon_{D}$ function in Equations (6)–(9) gives
(10)$$U_{p}=\{\begin{array}{c} C{\left(\frac{\rho_{0}}{\rho_{s}}\right)}^{2}\sigma_{y}\left(0.1\frac{v_{c}}{v_{w}}+1\right)\left(1-\gamma \frac{\rho_{0}}{\rho_{s}}\right)+\frac{1}{2}\rho_{0}v_{c}^{2}\;\;0<v_{c}<v_{w}\\ 1.1C{\left(\frac{\rho_{0}}{\rho_{s}}\right)}^{2}\sigma_{y}\left(1-\gamma \frac{\rho_{0}}{\rho_{s}}\right)+\frac{1}{2}\rho_{0}v_{c}^{2}\;v_{c}\geq v_{w} \end{array}$$

Substituting Equation (10) into Equations (7) and (8), the theoretical predictions of $U_{p}^{s}$ and ${\bar{U}}_{p}$ can thus be obtained.

By adopting the above model, the theoretical predictions of the specific plastic energy dissipation $U_{p}^{s}$ and the normalized plastic energy dissipation ${\bar{U}}_{p}$ at densification crushing are presented in Figure 13 with respect to the normalized crushing velocity parameter $\bar{V}=v_{c}/v_{w}$, respectively. To compare with the theoretical predictions, the two energy absorption indices calculated directly from the crushing simulations within the analyzed velocity and density range have also been presented in Figure 13. The comparison between the predictions and the FE results shows a satisfactory agreement, validating the effectiveness of theoretical model.

It is also found from Figure 13a that, for the analyzed relative density range, a consistent trend can be summarized from the curves as the specific plastic energy dissipation of higher relative density is obviously larger than that of lower relative density. Given the lightweight as design consideration, this conclusion can be an important reference in the engineering application of auxetic cellular structures. Differing from the specific plastic energy dissipation, when crushing at low velocities, the normalized plastic energy dissipation increases as the relative density is increased (Figure 13b). While at high crushing velocities, the structure relative density possesses negative effect on the capacity of energy absorption, demonstrating a shifting point at the intersection of the curves. So under high-velocity crushing, lower relative density results in higher normalized plastic energy dissipation during the crushing process and exhibits a higher rate of increase in energy dissipation capacity with an increasing crushing velocity. The shifting point for the proportional relation between the crushing velocity and the relative density is observed to be around $\bar{V}=1$, which actually represents the critical transition velocity between the transition mode and the dynamic mode. The results suggest that the crush hardening effect is more prominent for cellular structure with lower relative density under high-velocity crushing. To further validate the theoretical model, extensive experiments are suggested, which will be conducted in future work.

## 6. Conclusions

In the present work, a three-dimensional auxetic cellular structure consisting of re-entrant hexagonal cells has been proposed. A beam-based finite element method has been utilized to investigate the dynamic behaviors of the 3D re-entrant auxetic structure under direct crushing. Insights into the influence of crushing velocity and relative density on the crushing mode, crushing strength and energy absorption have been presented.

The detailed deformation process of the 3D re-entrant auxetic structure under the crushing velocity ranging from 2 m/s to 50 m/s has been described with the features of local densification and lateral contraction due to auxetic effect and three specific crushing modes have been observed, namely quasi-static, transition and dynamic mode. Further, the dependence of the crushing velocity and the relative density on the crushing mode has been identified, forming a mode classification map for the 3D re-entrant auxetic structure. The crushing strength of the 3D re-entrant auxetic structure reveals to increase with the increase in crushing velocity and relative density. Moreover, an analytical formula of dynamic plateau stress has been deduced by introducing the rate-dependent densification strain, which proves to present theoretical predictions agreeing well with simulation results. By establishing an analytical model, we have also studied the role of relative density on the energy absorption capacity of the 3D re-entrant auxetic structure with respect to crushing velocity. The theoretical and simulation results both show that the specific plastic energy dissipation of the structure increases consistently with the increase in the relative density, while the normalized plastic energy dissipation has an opposite sensitivity to the relative density when the crushing velocity exceeds the critical transition velocity.

The results in this work can be a preliminary complement to the dynamic analysis of the 3D re-entrant auxetic structure and provides an extensive reference for the crushing resistance design of the auxetic structure. Future work will focus on the experimental validation of the results and considering more factors in mechanical analysis such as the size effect and the matrix material properties.

## Figures and Tables

**Figure 1 materials-12-00460-f001:**
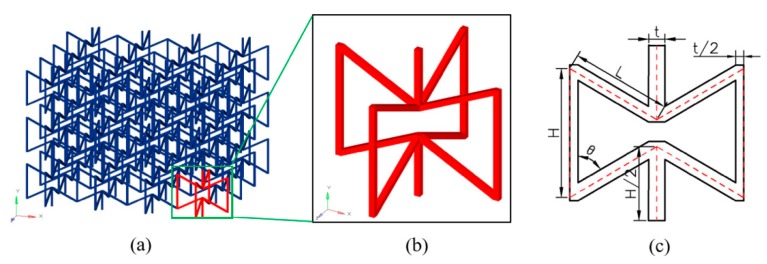
(**a**) 3D re-entrant auxetic cellular structure; (**b**) Representative volume element (RVE); (**c**) Geometrical features of the RVE.

**Figure 2 materials-12-00460-f002:**
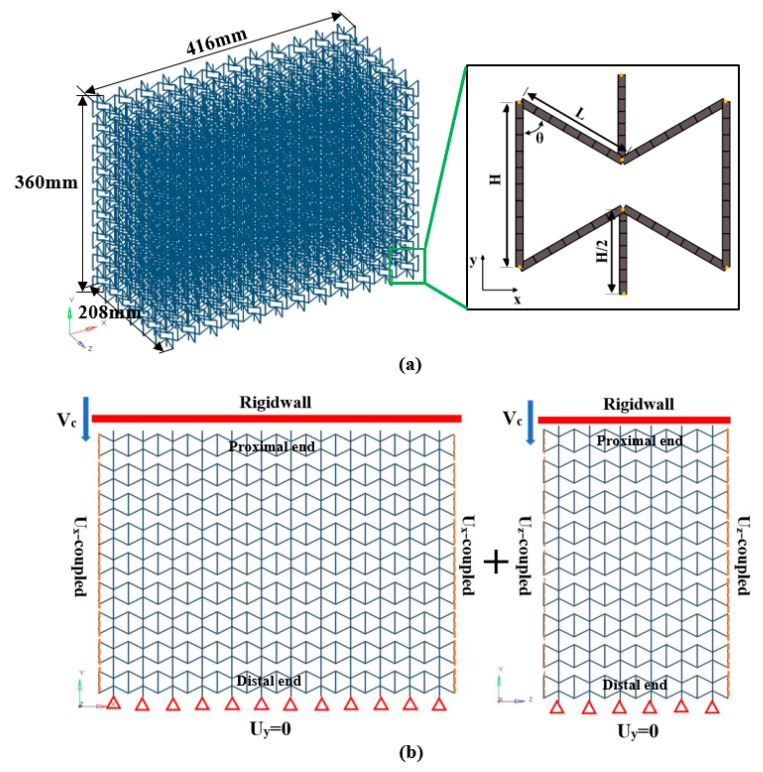
The numerical simulation model (12 × 11 × 6 type): (**a**) Overview of the mesh model; (**b**) The boundary conditions when crushing along the y direction.

**Figure 3 materials-12-00460-f003:**
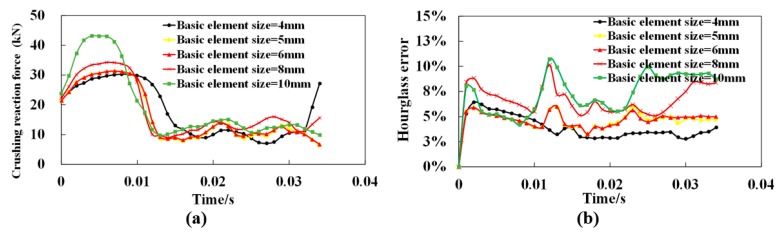
Convergence of the simulations under a constant velocity $v_{c}$ of 30 m/s with the basic element size varying from 4 mm to 10 mm: (**a**) Crushing reaction force; (**b**) Hourglass error.

**Figure 4 materials-12-00460-f004:**
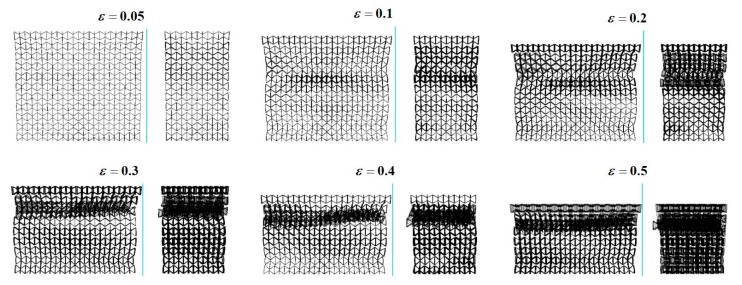
Deformation process of the 3D re-entrant auxetic structure under the crushing velocity $v_{c}$ = 5 m/s.

**Figure 5 materials-12-00460-f005:**
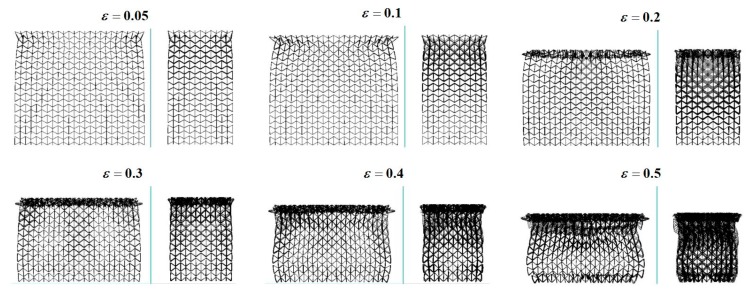
Deformation process of the 3D re-entrant auxetic structure under the crushing velocity $v_{c}$ = 15 m/s.

**Figure 6 materials-12-00460-f006:**
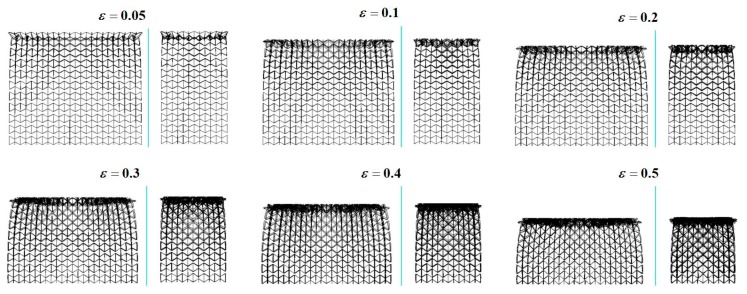
Deformation process of the 3D re-entrant auxetic structure under the crushing velocity $v_{c}$ = 50 m/s.

**Figure 7 materials-12-00460-f007:**
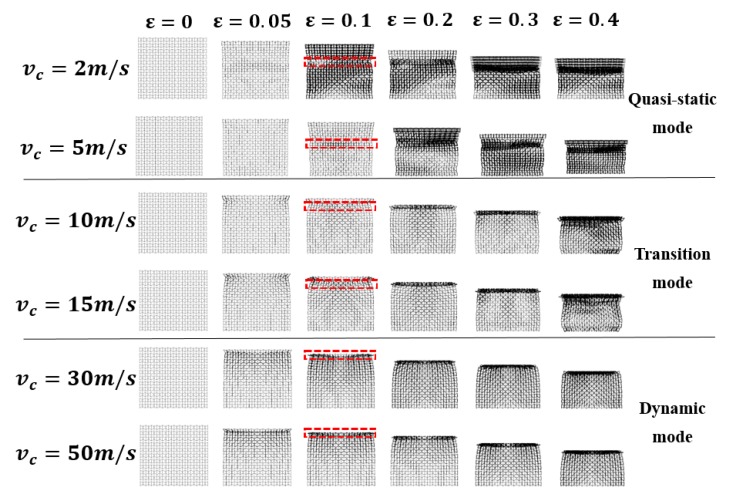
Deformation processes of the3D re-entrant auxetic structure ($\rho_{R}=1\%$) under different crushing velocities.

**Figure 8 materials-12-00460-f008:**
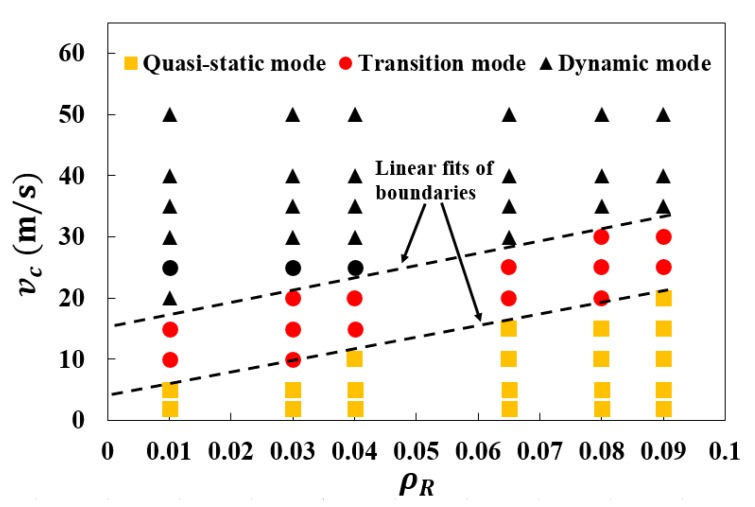
Mode classification map for the 3D re-entrant auxetic structures under dynamic crushing. The markers are the simulation results showing different crushing modes. The dashed lines represent the linear fits of boundaries associated with the transition among three specific crushing modes.

**Figure 9 materials-12-00460-f009:**
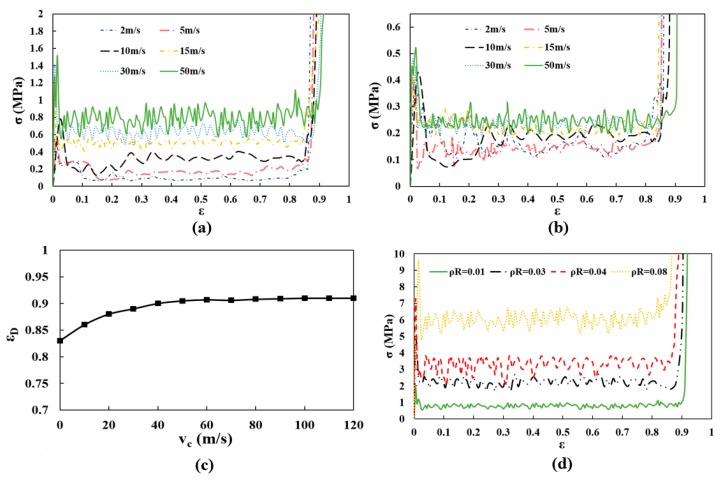
Dynamic crushing strength of the 3D re-entrant auxetic structure: (**a**) Crushing stresses at the proximal end; (**b**) Crushing stresses at the distal end; (**c**) Dynamic densification strains $\varepsilon_{D}$ under different crushing velocities; (**d**) Crushing stresses (proximal end) of structures with different relative densities.

**Figure 10 materials-12-00460-f010:**
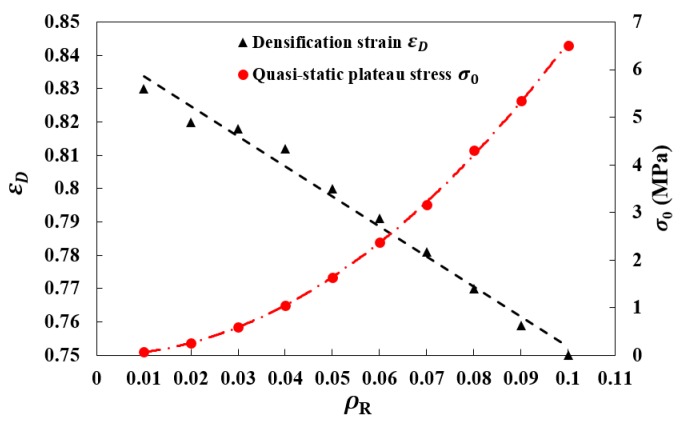
The quasi-static plateau stress $\sigma_{0}$ and the densification strain $\varepsilon_{D}$ with respect to relative density.

**Figure 11 materials-12-00460-f011:**
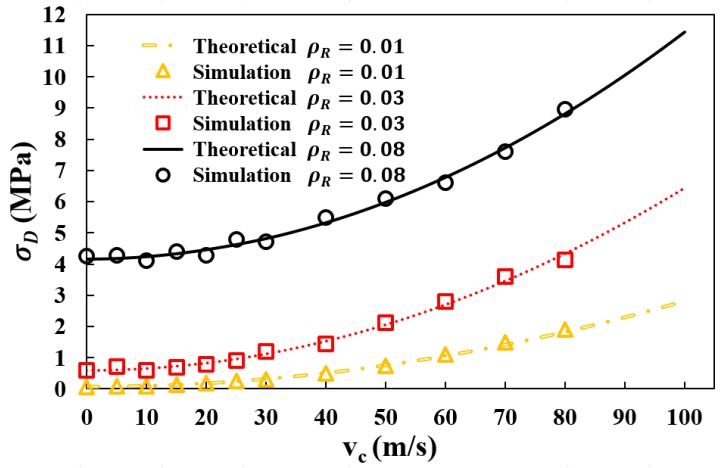
The Dynamic plateau stress $\sigma_{D}$ from numerical simulation and theoretical prediction using Equation (6).

**Figure 12 materials-12-00460-f012:**
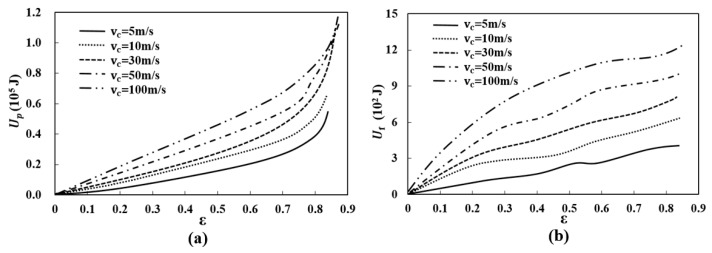
(**a**) Plastic energy dissipation $U_{p}$ versus the crushing strain ε; (**b**) Frictional energy dissipation $U_{f}$ versus the crushing strain ε.

**Figure 13 materials-12-00460-f013:**
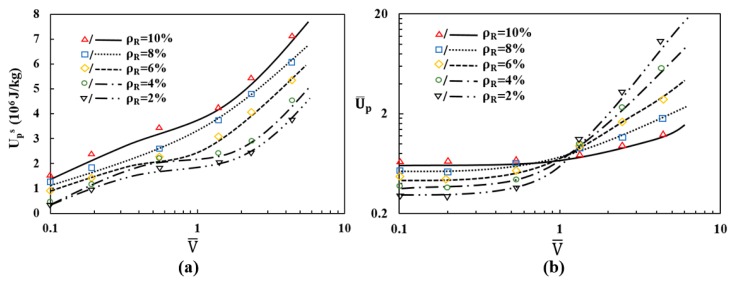
(**a**) Specific plastic energy dissipation $U_{p}^{s}$ versus the normalized crushing velocity $\bar{V}$; (**b**) Normalized plastic energy dissipation ${\bar{U}}_{p}$ versus the normalized crushing velocity $\bar{V}$.

**Table 1 materials-12-00460-t001:** Dynamic plateau stress $\sigma_{D}$ from numerical simulation and theoretical prediction using Equations (2)–(4).

Relative Density $\rho_{R}$	Crushing Velocity $v_{c}$ (m/s)	Dynamic Plateau Stress $\sigma_{D}\;$ (MPa)		
		Theoretical	Simulation	Deviation (%)
0.01	5	0.081	0.080	1.11
	15	0.145	0.142	2.03
	30	0.337	0.311	8.36
	50	0.804	0.733	9.66
	70	1.708	1.521	12.31
0.03	5	0.731	0.715	2.26
	15	0.718	0.692	3.81
	30	1.286	1.216	5.77
	50	2.294	2.127	7.86
	70	4.015	3.611	11.20
0.08	5	4.431	4.291	3.26
	15	4.616	4.398	4.96
	30	5.058	4.713	7.32
	50	6.716	6.112	9.88
	70	8.608	7.612	13.08

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
