# Peer review of "Dynamic Crushing Analysis of a Three-Dimensional Re-Entrant Auxetic Cellular Structure"

_materials, 2019, doi:10.3390/ma12030460_

Round 1
Reviewer 1 Report
Title: Dynamic crushing analysis of a three-dimensional re-entrant auxetic cellular structure
Authors: Tao Wang and colleagues
Overall assessment
In this manuscript, Sections 1 and 2 are well written and easy to understand. Nevertheless, it seems to me that the authors were confused in writing the sections after 3; therefore, it is often difficult to understand the simulation results appropriately. I strongly recommend the authors to revise the manuscript by careful and objective readings.
Specific comments
1. It is difficult to understand how to derive Eq. (1) in details; therefore, the derivation procedure should be demonstrated. Otherwise, the reference describing the procedure should be quoted.
2. The source of the elastic properties of PA6 should be denoted.
3. It is convenient to describe the dimensions of whole model, even though the model consists of 12 x 11 x 6 cells.
4. There are no definitions what “4-10 mm” listed in the upper right of Figs. 3(a) and (b) represent. These numbers seem to correspond to the “element sizes”, but it is difficult to understand what the “element size” corresponds to in Fig. 2(a). To clarify the element size, Figs. 2(a), 3(a), and 3(b) should be elaborated.
5. The label of the vertical axis of Fig. 3(b) should be revised to Hourglass error to reduce the misunderstandings of the readers.
6. It is necessary to demonstrate the velocity in the simulation shown in the caption of Fig. 3, isn’t it?
7. The authors describe as “the models with meshing size less than 6mm can achieve a controlled artificial strain energy in 5% of the total internal energy, indicating the hourglass error is acceptable”, but it is difficult to understand why the authors evaluate the validity. It is necessary to demonstrate the background of the evaluation.
8. It is difficult to identify the characters demonstrated at the upper right of the model, which are too small. If these characters are not necessary, they must be removed.
9. It is difficult to understand the differences among the three crushing modes in details. The authors must demonstrate these differences using a figure.
10. The definitions of the dashed lines in Fig. 8 are missing.
11. The definition of dynamic densification strain is missing; therefore, it is difficult to understand what Figs. 9(c) and 10 represents at all. This issue is also applicable to the dynamic plateau stress.
12. The locations of the figures are not often appropriate. If the figure appears before the corresponding descriptions, it is difficult to understand what the figure represents accurately. I strongly recommend the authors to show the figures after the corresponding explanations.
Recommendation
Major revisions are required and the authors must revise the manuscript while providing a point-by-point response to the comments. Unfortunately, the descriptions in the latter sections are often complicated. It seems to me that the authors were confused during the preparation of the manuscript. I strongly recommend the authors to revise the manuscript by careful and objective readings repeatedly.
Author Response
Response to Reviewer 1 Comments
Overall assessment
In this manuscript, Sections 1 and 2 are well written and easy to understand. Nevertheless, it seems to me that the authors were confused in writing the sections after 3; therefore, it is often difficult to understand the simulation results appropriately. I strongly recommend the authors to revise the manuscript by careful and objective readings.
Response: Thank you for your detailed review comments. The manuscript has been revised as suggested, and all changes made have been highlighted using yellow background.
Point 1: It is difficult to understand how to derive Eq. (1) in details; therefore, the derivation procedure should be demonstrated. Otherwise, the reference describing the procedure should be quoted.
Response 1: Thank you for the suggestion. The relative density of the 3D re-entrant auxetic structure, i.e. the ratio between the effective density of cellular structure and the density of strut material, can be obtained by dividing the volume occupied by all the struts and the total volume. A detailed derivation procedure has been added in section 2.1 and highlighted using yellow background in the revised manuscript to make Eq. (1) more understandable.
Point 2: The source of the elastic properties of PA6 should be denoted.
Response 2: Thank you for the suggestion. The elastic properties of PA6 have been adopted by referring to the paper “Scaffaro R, Maio A, Tito A C. High performance PA6/CNTs nanohybrid fibers prepared in the melt[J]. Composites Science & Technology, 2012, 72(15):1918-1923.”. The quotation of this paper has been add as [23] reference and highlighted using yellow background in section 2.2.
Point 3: It is convenient to describe the dimensions of whole model, even though the model consists of 12 x 11 x 6 cells.
Response 3: Thank you for the suggestion. The dimensions of whole model have been added in section 2.2 as suggested.
Point 4: There are no definitions what “4-10 mm” listed in the upper right of Figs. 3(a) and (b) represent. These numbers seem to correspond to the “element sizes”, but it is difficult to understand what the “element size” corresponds to in Fig. 2(a). To clarify the element size, Figs. 2(a), 3(a), and 3(b) should be elaborated.
Response 4: Sorry for the blurred description of this section. The numbers “4-10 mm” listed in the upper right of Figs. 3(a) do correspond to the “element sizes”. More specifically, these numbers represent the basic element size in meshing the strut with the two-node beam element BEAM188 using Abaqus® automatic mesh technique.
A detailed discussion in this section has been added as suggested to make the description about “element size” clearer. Also, the legends in Figure 3(a) and (b) have been improved.
Point 5: The label of the vertical axis of Fig. 3(b) should be revised to Hourglass error to reduce the misunderstandings of the readers.
Response 5: Thank you for the suggestion. The label of the vertical axis of Fig. 3(b) has been revised to Hourglass error as suggested.
Point 6: It is necessary to demonstrate the velocity in the simulation shown in the caption of Fig. 3, isn’t it?
Response 6: Thank you for the suggestion. To demonstrate the velocity in the simulations for mesh sensitivity investigation, the caption of Figure 3 has been revised to “Figure 3. Convergence of the simulations under a constant velocity vc of 30m/s with the basic element size varying from 4mm to 10mm: (a) Crushing reaction force; (b) Hourglass error”.
A detailed discussion about the mesh sensitivity investigation has also been added as suggested and highlighted using yellow background in section 2.2.
Point 7: The authors describe as “the models with meshing size less than 6mm can achieve a controlled artificial strain energy in 5% of the total internal energy, indicating the hourglass error is acceptable”, but it is difficult to understand why the authors evaluate the validity. It is necessary to demonstrate the background of the evaluation.
Response 7: Thank you for the suggestion. Abaqus explicit analysis used in this work uses hourglass control during simulations. If the artificial strain energy is greater than approximately 5% of the total strain energy, i.e. hourglass error exceeds 5%, hourglassing could be a problem and should be further analysed by changing mesh and elements to see weather results correlate. This criterion is adopted by referring to the Abaqus/CAE User’s Manual.
A detailed discussion about the background of the evaluation has been added in section 2.2 as suggested. The Abaqus manual has also been referred for better evaluation.
Point 8: It is difficult to identify the characters demonstrated at the upper right of the model, which are too small. If these characters are not necessary, they must be removed.
Response 8: Thank you for the suggestion. Those characters in Fig. 7 show the project titles for the simulations, and are not necessary here. They have been removed as suggested. All the figures have been improved to remove the meaningless characters in the revised manuscript.
Point 9: It is difficult to understand the differences among the three crushing modes in details. The authors must demonstrate these differences using a figure.
Response 9: Thank you for the suggestion. Differing from the crushing patterns reported for regular hexagonal honeycombs as typical ‘X’, ‘V’ and ‘I’ types, no obvious shape can be characterized for the deformation mode of auxetic structures in this work. As depicted in Figure 7, the main differences among the three crushing modes lie in the location of initial collapse band and the growth process of densification core. For better illustration, the features of three specific crushing modes have been identified in Figure 7.
Point 10: The definitions of the dashed lines in Fig. 8 are missing.
Response 10: Sorry for missing definition. The dashed lines represent the linear fits of boundaries associated with the transition among three specific crushing modes. In the revised manuscript, a marker for dashed line has been labeled in Figure 8, and a further description has also been added in the corresponding caption as suggested.
Point 11: The definition of dynamic densification strain is missing; therefore, it is difficult to understand what Figs. 9(c) and 10 represents at all. This issue is also applicable to the dynamic plateau stress.
Response 11: Densification strain is defined as a strain at which densification begins. However, normally there is no abrupt transition from plateau regime to densification regime. Further, as the recent studies have indicated an impact velocity dependent densification strain as the crushing stress in cellular structures, the concept of a dynamic densification strain is therefore adopted in this work. By referring to the publication [35], the dynamic densification strain εD is defined as the intersection of the tangents to the plateau and densification regimes in this study, and the value is then determined numerically from the obtained stress-strain curves within FE simulation.
In this study, by referring to the publication [30], the dynamic plateau stress is defined as the average value of compression stresses for a compressive strain of between the 0.5% strain and the dynamic densification strain εD, and the theoretical expression for the dynamic plateau stress is given in Eq. (2).
The detailed definitions of dynamic densification strain and dynamic plateau stress have been added as suggested to make the work in section 4 clearer.
Point 12: The locations of the figures are not often appropriate. If the figure appears before the corresponding descriptions, it is difficult to understand what the figure represents accurately. I strongly recommend the authors to show the figures after the corresponding explanations.
Response 12: Thank you for the suggestion. To make the figures more understandable, the locations of the figures have been revised as suggested.
Thank you for your professional reviews and suggestions.

Reviewer 2 Report
Review of the paper entitled "Dynamic crushing analysis of a three-dimensional re-entrant auxetic cellular structure" submitted for publication to Materials.
I found the paper interesting, the topic of the manuscript itself is relevant and the study can have a remarkable scientific soundness.
However even if the reviewer considers positively the vast numerical work done, in order for this paper to be published a deep general restyle of the manuscript is required.
In particular the reviewer found the Figures and all the illustrations not enough good for an international journal in fact the annotation are often blurred and very difficult to read, the axis' labels are very small and even the general purpose of several pictures is not easy understandable due to the low level of details.
The english often is not perfect/polished as required by an international journal, therefore an extensive modification of the text could also be beneficial.
I suggest a major revision in order to fix these fundamental points.
Best.
Author Response
Response to Reviewer 2 Comments
Overall assessment
I found the paper interesting, the topic of the manuscript itself is relevant and the study can have a remarkable scientific soundness.
However even if the reviewer considers positively the vast numerical work done, in order for this paper to be published a deep general restyle of the manuscript is required.
In particular the reviewer found the Figures and all the illustrations not enough good for an international journal in fact the annotation are often blurred and very difficult to read, the axis' labels are very small and even the general purpose of several pictures is not easy understandable due to the low level of details.
The english often is not perfect/polished as required by an international journal, therefore an extensive modification of the text could also be beneficial.
Response: Thank you for your detailed review comments. The manuscript has been revised as suggested, and all changes made have been highlighted using yellow background.
All the figures and the illustrations have been improved to be clearer, and some meaningless characters in several figures have been removed.
The english has been polished to improve the paper’s quality.
Thank you for your professional reviews and suggestions.

Reviewer 3 Report
The article is devoted to very interesting topic i.e dynamic behavior of auxetic structure subjected to crushing. The analyzed structure is typical re-entrant geometry which is known of exhibiting negative Poisson’s ratio. Considering the fact that there are not many articles devoted to dynamics of auxetic I think this on is a valuable contribution to this topic. However, there are some quite significant publications that concerns the same or very similar problems of dynamics of auxetic structures and has not been mentioned by authors including several works by F. Scarpa e.g.:
1. Dynamic properties of high structural integrity auxetic open cell foam (doi:10.1088/0964-1726/13/1/006)
2. Dynamic crushing of auxetic open-cell polyurethane foam (doi: 10.1243/095440602321029382)
Also, examplary articles that has just been recently published:
3. “Computational Analysis of the Mechanical Impedance of the Sandwich Beam with Auxetic Metal Foam Core” (10.1002/pssb.201800423)
4. “Crushing Behavior of Graded Auxetic Structures Built from Inverted Tetrapods under Impact” 10.1002/pssb.201800040
When it comes to the results obtained, the simulations along with its detailed analysis is very comprehensive and I think it does not require any significant improvements. Also analytical model of dynamic plateau stress and energy absorption capacity is sufficient. However, I would greatly appreciate any experimental results in order to validate numerical ones. This is the main weakness of this work form my point of view. Nonetheless, I think it deserves being published .
Author Response
Response to Reviewer 3 Comments
Overall assessment
The article is devoted to very interesting topic i.e dynamic behavior of auxetic structure subjected to crushing. The analyzed structure is typical re-entrant geometry which is known of exhibiting negative Poisson’s ratio. Considering the fact that there are not many articles devoted to dynamics of auxetic I think this on is a valuable contribution to this topic. However, there are some quite significant publications that concerns the same or very similar problems of dynamics of auxetic structures and has not been mentioned by authors including several works by F. Scarpa e.g.:
1. Dynamic properties of high structural integrity auxetic open cell foam (doi:10.1088/0964-1726/13/1/006)
2. Dynamic crushing of auxetic open-cell polyurethane foam (doi: 10.1243/095440602321029382)
Also, examplary articles that has just been recently published:
3. “Computational Analysis of the Mechanical Impedance of the Sandwich Beam with Auxetic Metal Foam Core” (10.1002/pssb.201800423)
4. “Crushing Behavior of Graded Auxetic Structures Built from Inverted Tetrapods under Impact” 10.1002/pssb.201800040
When it comes to the results obtained, the simulations along with its detailed analysis is very comprehensive and I think it does not require any significant improvements. Also analytical model of dynamic plateau stress and energy absorption capacity is sufficient. However, I would greatly appreciate any experimental results in order to validate numerical ones. This is the main weakness of this work form my point of view. Nonetheless, I think it deserves being published.
Response: Thank you for your detailed review comments. The manuscript has been revised as suggested, and all changes made have been highlighted using yellow background.
We have read the suggested publications carefully and received many useful information from them. Two of the publications have been quoted as [3] and [5] references in the revised manuscript to further support the claim in our work.
To further validate the theoretical model, extensive experiments are preferred. Since the prototyping of the proposed structure is in progress, the validation experiments will be conducted in future work.
Thank you for your professional reviews and suggestions.

Round 2
Reviewer 1 Report
The revisions are adequately conducted; therefore, I'd like the revised version to be published in the journal as it is.
Reviewer 2 Report
The author has remarkably improved the paper by adding some meaningful parts and amendments to better explain the main goals achieved and the purposes of the study.
The reviewer in particular has appreciated the improvements made in all the illustrations.
After a careful revision of the updated version of the manuscript, and following the previous positive opinion I gave about the original paper, I can recommend the publication in Materials.
This manuscript is a resubmission of an earlier submission. The following is a list of the peer review reports and author responses from that submission.